# Benefits under the Sea: The Role of Marine Compounds in Neurodegenerative Disorders

**DOI:** 10.3390/md19010024

**Published:** 2021-01-08

**Authors:** Mariano Catanesi, Giulia Caioni, Vanessa Castelli, Elisabetta Benedetti, Michele d’Angelo, Annamaria Cimini

**Affiliations:** 1Department of Life, Health and Environmental Sciences, University of L’Aquila, 67100 L’Aquila, AQ, Italy; mariano.catanesi@guest.univaq.it (M.C.); giulia.caioni@guest.univaq.it (G.C.); vanessa.castelli@univaq.it (V.C.); elisabetta.benedetti@univaq.it (E.B.); 2Sbarro Institute for Cancer Research and Molecular Medicine, Department of Biology, Temple University, Philadelphia, PA 19122, USA

**Keywords:** marine drugs, neurodegenerative diseases, Parkinson’s disease, Alzheimer’s disease, brain, antioxidants

## Abstract

Marine habitats offer a rich reservoir of new bioactive compounds with great pharmaceutical potential; the variety of these molecules is unique, and its production is favored by the chemical and physical conditions of the sea. It is known that marine organisms can synthesize bioactive molecules to survive from atypical environmental conditions, such as oxidative stress, photodynamic damage, and extreme temperature. Recent evidence proposed a beneficial role of these compounds for human health. In particular, xanthines, bryostatin, and 11-dehydrosinulariolide displayed encouraging neuroprotective effects in neurodegenerative disorders. This review will focus on the most promising marine drugs’ neuroprotective potential for neurodegenerative disorders, such as Parkinson’s and Alzheimer’s diseases. We will describe these marine compounds’ potential as adjuvant therapies for neurodegenerative diseases, based on their antioxidant, anti-inflammatory, and anti-apoptotic properties.

## 1. Introduction

Throughout history, nature and medicine have shown a strong relation, as highlighted by the wide use of therapeutic biomolecules in traditional medicines for thousands of years. During the ancient Greece and early Byzantium periods, the therapeutic application of marine organisms was deeply rooted in Mediterranean populations. In particular, the use of marine invertebrates gained such importance in medical practice that many works were dedicated to them. Classical texts described the pharmaceutical or therapeutical properties and the way of administration and the manipulation of the raw materials. For example, invertebrates were used in the forms of juices, beverages, pulverized products, broth, unguent, or eaten as fresh or dry flesh [1]. A considerable contribution to marine drugs research was also given by traditional Chinese medicine. The use of marine herbs and marine herbal formulas belongs to a thousand-year tradition, and the in-depth knowledge of marine drugs and other organisms was enriched by means of local chronicles, folk formulas, monographs, first medical prescriptions, and dietary suggestions. However, careful discrimination occurred only with the advent of the scientific method and scientists’ dedication. All of this information and new discoveries were collected in the Chinese marine materia medica. It is a kind of encyclopedia, and it represents the best compendium about cyanobacteria, algae, marine animals, and minerals. Chinese marine materia medica is also considered the starting point for the development of new market drugs [2]. This long tradition demonstrates that the interest in marine products developed in the ancient world, even if only current studies and the use of modern technologies and have given a lot of insight into the molecules involved in the beneficial effects. Nowadays, the development of appropriate equipment has allowed a better exploration of marine ecosystems, leading to the enrichment of knowledge in the field of aquatic organisms. The access to unknown ecological niches has given the opportunity to came across new marine compounds and to study their possible use [3]. However, this alone does not justify the progress in marine medicine. In fact, a considerable support comes from chemistry and modern molecular biology, which together point towards an innovative approach, considering the production of synthetic analogs and the engineering manipulation of marine molecules [3]. In particular, the application of genome sequencing techniques helped the comprehension of mechanisms under the biosynthetic pathways, also allowing the cloning of particular compounds, as the case of the actinomycetes *Salinispora tropica.* The accurate analysis determined the identification of secondary metabolites, showing the importance of bioinformatic skills in addition to traditional biology [4]. The neuroscience research converted the ancient medicaments into the opportunity of developing neuroprotective drugs for the treatment of neurodegenerative disorders. Despite 250 years of marine research, around 91% of the species present in the sea still lack a detailed description. Since the first marine organisms appeared about 3500 million years ago, they had the opportunity to develop different mechanisms to survive in the unfriendly ambient; these adverse conditions likely favored the development of such a large number of bioactive molecules for counteract environmental stress. In fact, to survive in extreme habitats or against predators, marine species developed secondary metabolic pathways to produce molecules to adjust their lifestyles. Indeed, despite the environmental change, marine organisms do not experience severe oxidative and photodynamic damages because it is known that marine organisms can synthesize bioactive molecules to protect themselves from external factors.

Moreover, the production of bioactive compounds has an ecological meaning, since they allow the survival and the adaptation in the marine environment. Defense strategies include the release of harmful toxins, especially by benthonic species, such as algae or mollusks [5]. For example, heterobranch mollusks produce a wide range of molecules against predators and all nudibranchs use anti-fouling molecules [6]. Flatfish, such as Moses sole (*Pardachirus marmoratus*), produce defense secretions, containing anti-feeding and shark-repellent agents [7]. Other examples are the production of tetrodotoxin by several pufferfish species or conotoxins, isolated from venom of the marine cone snails. However, secondary metabolites can also attract organisms for a reproductive purpose or induce attachment and metamorphosis of larvae produced by sessile organisms [8]. On the basis on the evidence reported above, marine organisms attracted scientists’ interest for the potential biological activities of their primary and secondary metabolites. Terpenes, shikimates, polyketides, acetogenins, peptides, alkaloids, and many uncharacterized structures, extracted and purified from marine resources [9], showed various pharmacological activities as antioxidant [10], antibacterial [11], anticancer [12], antiviral [13], anti-inflammatory [14], antidiabetic [15], antihypertensive [16], and anticoagulant [17]. This review reports the main marine products and their derivatives, describing their antioxidant, anti-inflammatory, and neuroprotective properties. Finally, a focus on their therapeutic potential as adjuvant agents to gold standard therapies in neurodegenerative diseases, such as Parkinson’s (PD) and Alzheimer’s diseases (AD), is reported.

## 2. Neurodegenerative Diseases

Neurodegenerative diseases are a heterogeneous group of late-onset disorders caused by the progressive dysfunction and death of neuronal cells, leading to a series of cognitive and movement disorders. The incidence of these diseases increases with age and is expected to become more common due to extended life expectancy. Aging is considered the main factor in neurodegenerative processes. Neurodegeneration is characterized by the neuronal loss of function, death, and aggregation of misfolded proteins, and the formation of intracellular and extracellular deposits. Among the key features of neurodegenerative diseases, the excessive production of reactive oxygen species (ROS) and inflammation play an important role, representing the direct consequences of perturbation in central nervous system (CNS) homeostasis [18]. Nucleic acid oxidation leads to the formation of 8-hydroxydeoxyguanosine in mitochondrial and nuclear DNA. These modifications can be evaluated in terms of DNA strand breakage. It is not surprising that AD patients have an increased level of breaks in the cerebral cortex than controls [19]. Moreover, the degeneration of dopaminergic neurons contributes to oxidative stress and several markers supporting this correlation. In AD patients, the presence of amyloid plaques (Aβ) seems to be related to an increase in oxidative stress, determining an impairment in energy production mediated by mitochondria. In particular, Aβ is a metalloprotein, and it has binding sites for transition metals in N-terminal sequence [20]. The ability to bind copper and iron ions results in alterations in metals’ oxidation state, producing hydrogen peroxide, which is responsible for the increase in oxidative stress [21]. In PD patients, the excessive ROS amount has been related to a reduced activity in mitochondrial complex I of *substantia nigra pars compacta* dopaminergic neurons [22] and many studies suggest that oxidative stress may be related to dopamine (DA) metabolism. For example, the product from DA oxidation DA-quinone can modify several molecules, such as glutathione (GSH) and proteins, including parkin, α-synuclein, ubiquitin carboxy-terminal hydrolase L1, and protein/nuclease acid-deglycase [23]. Many in vitro and in vivo studies showed that the impairment in parkin ubiquitin ligase activity is the consequence of oxidative stress [24,25]. In particular, the conditions associated with ROS generation also promote α-synuclein aggregation, suggesting that oxidative reactions may be critical in forming Lewy bodies [26]. Reactive species are regulators of signaling pathways, activating several transcription factors, such as nuclear factor-kappa B (NF-κB) or activator protein-1, which, in turn, regulate the expression of several genes, such as adhesion molecules, pro-inflammatory cytokines, growth factors, inducible nitric oxide synthase, cyclooxygenase-2 and cytosolic phospholipases A2 [27]. The activation of NF-κB has been shown in both neurons and astrocytes of AD patient brains. In particular, the neuronal stimulation of NF-κB determines the activation of anti-oxidant enzymes, such as superoxide dismutase 2, while in astrocytes and microglia, there is an increase in pro-oxidants. In vitro studies demonstrated that the exposure to fibrillary Aβ stimulates microglia to produce pro-inflammatory cytokines, such as interleukin 6/1β, transforming growth factor-β, chemokines, tumor necrosis factor-α, and macrophage-stimulating factors. Although inflammation represents a protective response, the persistence of the inflammatory state has adverse effects on tissue. The consequences of neuroinflammation are not only related to cell death, which is the final event, but they can also cause a series of events that precede neuronal apoptosis. First of all, the synaptic dysfunction, which includes impairment in signal transmission and loss of synaptic activity. The cognitive function, short- and long-term memory impaired, and the causes have been identified in pro-inflammatory mediators. For example, Mishra et al. showed that the in vitro administration of inflammatory cytokine such as interleukin 1β induces a synaptic loss in rat hippocampal neurons. The mechanism reported could involve the interleukin-1β-mediated up-regulation of cyclooxygenase-2 in neuronal cells and astrocytes [28]. Several in vitro studies demonstrated the activity of protection against Aβ- and 6-hydroxydopamine (6-OHDA)-induced neurotoxicity of steroids and 11-dehydrosinulariolide extracted from soft corals of *Sinularia. genus.* [29]. In the following paragraphs, the classification for principal marine drugs and related effects on CNS system are reported. Moreover, we will also describe the various methods of isolation and extraction for the active biomolecules and finally we will describe the most promising compounds for the treatment of PD and AD.

## 3. Classification of Marine Compounds

Marine compounds can be classified based on their chemical structures, and they include amino acids, simple peptides, nucleotides, lipids, polysaccharides, cytokinin, alkaloids, toxins, steroids, prostaglandins, etc. Since the number of bioactive compounds is broad, only a part of these will be considered. Table 1 summarizes the main neuroprotective substances, subdivided into five groups, based on their chemical properties; moreover, the marine sources and the reported effects are indicated.

After listing some of the molecules of neurobiological interest, it is necessary to focus on the main mechanisms through which they exert their beneficial effects. Given the high number and complexity of these pathways, only antioxidant, anti-inflammatory, and anti-apoptotic effects will be considered.

### 3.1. Principal Methods of Extraction, Separation, Isolation, and Identification

Since beneficial effects are not related to the entire organism, but only to the presence of defined molecules, efficient extraction methods are necessary. Many methods are currently used, allowing to obtain an extract, which undergoes other steps of manipulation. The next steps refer to separation and isolation techniques, which include many types of chromatography and micro- or nano-filtration. Then, the structure is usually analyzed using spectroscopy techniques (2D NMR, mass spectroscopy), and finally, the compound is evaluated in terms of biological, toxicological, and clinical effects [62]. This phase is fundamental to identify the anti-inflammatory, anti-bacterial, analgesic, and other activities, and the tests can be performed using in vitro or in vivo models.

These protocols have been adopted and modified to isolate many different kinds of compounds and many efforts were put in their improvement. Based on the recent advancements, conventional and non-conventional techniques can be distinguished. The conventional ones include decoction, maceration, percolation, infusion, reflux extraction, Soxhlet extraction. Hydro-distillation, ultrasound-assisted extraction, pressurized solvent extraction, enzyme-assisted extraction, and supercritical-fluid extraction are examples of non-conventional methods [63,64]. They are used in the extraction of carotenoids, such as fucoxanthin (FX) [65], sulfated polysaccharides, including fucoidan [66] and astaxanthin (AXT) [67]. This evidence suggests that there is no definitive method of extraction. The point is to adopt the best solution based on the molecules’ chemical properties, even if the difficulty is usually related to the low concentrations of active compounds in the extract. The use of alternative methods allows reducing time of extraction and the consumption of materials without influencing the quality of extracts. The substances should be evaluated in terms of biological, toxicological, and clinical effects [62]. This phase is fundamental to identify the anti-inflammatory, anti-bacterial, analgesic, and other activities, and the tests can be performed using in vitro or in vivo models. However, the study of structure-based design and the use of in silico models allow a complete characterization of the molecules, also in terms of their ability to bind particular substrates and cross the cellular membrane. The assessment of cytotoxicity may occur by means of MTS and MTT colorimetric assays, which are used as an indicator of cell viability. Among the preliminary studies on biological activity, the anti-proliferative activity assay and anti-oxidant capability may be evaluated [68]. Bacterial reverse mutation assay is performed to assess whether a given chemical can cause mutations in the DNA of bacteria, restoring its ability to synthesize a particular amino acid [69]. Moreover, it can be predicted their eventual role in inducing structural chromosomal abnormalities by means of chromosomal aberration assay. Mammalian in vivo tests, such as the mouse micronucleus assay and toxicological studies, deepen the interaction of these substances with complex organisms.

### 3.2. Anti-Oxidant, Anti-Inflammatory, and Anti-Apoptotic Effects: How do Marine Drugs Help US?

Marine drugs can exert protective effects, based on their anti-oxidant, anti-inflammatory, and anti-apoptotic properties. As mentioned above, ROS and oxidative stress are implicated in the onset of neurodegeneration. Although cells are provided with anti-oxidant defenses, neurons result vulnerable in long-term oxidative conditions. The consequences are more or less severe in accordance with the severity of the damage. Among the possible therapeutic strategies, anti-oxidant use appears to be promising, even if the efficacy of their administration in real patients is controversial [70,71].

Promising results derive from the study of natural marine carotenoids, for example. The interest in this class of compounds originates from several studies, which showed the beneficial effects of a dietary supplementation a mixture of natural carotenoids [72,73]. Carotenoids are terpenoids and naturally occur in archaea, plants, fungi, algae. They are responsible for the red, orange, or yellow color of organisms. The oxygenated forms of carotenoids, known as xanthophylls, include AXT, FX, zeaxanthin, neoxanthin, violaxanthin, etc., which are produced mainly by green microalgae, brown algae, and diatoms [74]. Animals cannot synthesize carotenoids, but they accumulate them through food. Many invertebrates can accumulate these molecules. For example, sponges, mollusks, crustaceans or tunicates, and echinoderms [75]. Carotenoids have a key role in scavenging free radicals and reactive species. Wu et al. demonstrated that the oral administration to rats leads to the recovery of glutathione peroxidase and superoxide dismutase activity and the increase in GSH levels. AXT can ameliorate rat brain aging by means of brain-derived neurotrophic factor (BDNF) upregulation; mature BDNF plays an essential role in memory [76] formation and storage and is downregulated in the brains of AD patients and other neuropathology. Moreover, decreased levels of malondialdehyde along with protein carbonylation and cyclooxigenase-2 expression were found [77]. Moreover, it can negatively influence microglia-dependent inflammatory responses, leading to the attenuation of tissue damage [78]. Other studies focused on the neuroprotective effect on cerebral ischemia-reperfusion damage in rats. The pre-treatment with AXT before inducing the ischemic injury resulted in being effective thanks to its anti-oxidant properties [79]. The protective effects have also been reported on human health. AXT can ameliorate brain function: the daily administration for a certain period (12 weeks) to healthy Japanese adults, feeling memory decline, gave positive results since the subjects demonstrated an improvement in composite and verbal memory [80]. FX shares some characteristics with AXT, including the anti-oxidant and the anti-inflammatory effects. It acts as a scavenger compound against organic free radicals, such as 1,1-diphenyl-2-picrylhydrazyl, 12-doxyl-stearic acid, radical adducts of nitroso benzene, and typical reactive species, including hydroxyl radical, singlet oxygen, and superoxide anion [81]. Moreover, the ability to inhibit intracellular ROS avoids the formation of DNA oxidation products and apoptosis hydrogen peroxide-mediated. The increase in catalase levels accompanies these protective actions [82]. Other properties have also been evaluated in vivo experiments, which showed that its administration as a dietary supplement determines a decrease in oxidative stress risk in high-fat diet-fed rats [83]. Regarding the neuroprotective effects, FX can enhance neuron survival in traumatic brain injury models. In particular, its administration in traumatic brain injury mice can ameliorate neurological deficits and tissue lesions, reducing oxidative stress. Malondialdehyde levels resulted decreased, while glutathione peroxidase was activated. However, the neuroprotection was lacking in traumatic brain injury nuclear factor E2-related factor 2-knockout mice, suggesting nuclear factor E2-related factor 2-dependant pathways in FX-activated mechanisms [41]. Neuroinflammation is a typical trait of CNS pathologies involving microglial cells pro-inflammatory mediators, a subset of enzymes, such as cyclooxygenase -1 and -2, nitro oxygenase, and other cytokines. Several compounds isolated from *Sinularia* genus of soft corals exhibit anti-inflammatory properties. In vivo studies on carrageenan-induced inflammation, rat models demonstrated that the administration of the cembranolide diterpene sinularin, isolated from *Sinularia querciformis* can inhibit microglial and astrocyte activity, with a decrease in inducible nitric oxide synthase levels and other inflammation markers in the dorsal horn of the lumbar spinal cord. Leukocyte infiltration and edema in the paw resulted ameliorated. Sinularin properties have also been evaluated In vitro, in lipopolysaccharide-stimulated RAW 264.7 cells, which are murine macrophages. It inhibits up-regulation of in inducible nitric oxide synthase and cyclooxygenase-2, promoting the production of tumor grow factor β protein [84]. Moreover, 11-Dehydrosinulariolide is isolated from *Sinularia flexibilis* and showed anti-apoptotic effects, inhibiting 6-OHDA- induced caspase-3/7 and NF-κB activation in SH-SY5Y cells [55]. HTP-1, a peptide derived from the sea-horse *Hippocampus trimaculatus*, showed a protective effect in in vitro models of AD. In particular, PC12 cells have been co-cultivated with BV2 cells (murine microglial cells) stimulated by Aβ_42_ oligomer, and HTP-1 showed to be protective towards PC12 cells. It can activate (via transforming growth factor-β) PI3K/Akt signaling pathway, which is known to promote survival [85]. The inhibition of neuronal death belongs to neuroprotective mechanisms, which have also been identified in marine algae. Fucoidan, isolated from *Undaria pinnatifida*, can reduce apoptosis in PC12 cells exposed to Aβ_25–35_ and D-galactose. It was demonstrated its role in improving PC12 viability, upregulating the expression of X-linked apoptosis inhibitor protein and other anti-apoptotic proteins. Cleaved caspase-3, caspase-8, and caspase-9 levels resulted decreased along with cytochrome c content in the cytoplasm [86]. Carotenoids are known to counteract oxidative stress, as described above, but they also can promote cell survival, activating several pathways [87]. Microalgae use pigment molecules to capture solar light, which is fundamental for photosynthetic processes. These pigments include zeaxanthin, AXT, lutein, canthaxanthin. For example, lutein protects neurons, avoiding Bcl-2-associated X protein accumulation, the activation of caspases, and loss of anti-apoptotic proteins in models of neurodegenerative diseases and cerebral ischemia [88]. Another example of neuroprotection is given by Xyloketal B, isolated from mangrove fungus *Xyloketal sp*. The interest in this compound derives from its anti-oxidative and anti-apoptotic properties, which make it a potential drug for the treatment of neurodegenerative diseases. It can control anti-apoptotic/pro-apoptotic ratio, preventing mitochondrial impairment and apoptosis. It showed the neuroprotective activity in ischemic brain injury. Xyloketal resulted in being effective in reducing levels of Bcl-2-associated X protein and cleaved caspase-3, and increasing levels of B-cell lymphoma-2 protein [89]. These are just some examples of molecules that exert neuroprotective effects: they play a role in counteracting oxidative stress, inflammation, and cell death. Each compound has a specific subset of characteristics and can activate multiple pathways. Because of their multitude, it is impossible to summarize or make a comprehensive classification of the compounds. For these reasons, only the bioactive marine compound and related studies that could be employed in AD and PD treatment will be described specifically in the next paragraphs.

### 3.3. The Effects of Marine Compounds on CNS

Marine molecules are highly heterogeneous, mainly due to the ocean’s coverage of about 70% of the surface of the earth [90] hosting a wide ecological, chemical, and biological diversity. This diversity in marine habitat gives marine molecules a large spectrum of action, attracting pharmaceutical researchers’ interest. The specific habitat where an organism grows influences the chemical nature of the marine primary and secondary metabolites. Through close cooperation between pharmaceutical industries and academic partners, it is possible to successfully collect, isolate and classify marine organisms, such as bacteria, fungi, micro-and macroalgae, cyanobacteria, and marine invertebrates from the seas. Extracts and purified compounds of these organisms can be studied for different therapeutically and biological activities; usually, over 60% of the pharmaceutical formulations’ active principles are natural products [91] or their synthetic derivatives or mimetics. Based on these observations, several biotechnologies research projects have been initiated, such as the Horizon 2020 (2020–in progress) project exploring different bioactive marine compounds or the SeaBioTech project, which harnesses marine potential microbes for industrial biotechnology (2012–2016). Therefore, it becomes crucial to use these molecules as potentially useful medicines against various types of diseases such as cancer, hypertension, diabetes, or neurodegenerative diseases. The approval of Prialt^®^ showed that the interest in marine drugs was not an idealistic purpose, but a real possibility to expand therapeutic strategies.

The selective blocker of N-type calcium channels Prialt^®^ (ziconotide) is the synthetic form of the peptide ω-conotoxin MVIIA [92], found in the sea snail’s venom *Conus Magus.* It is used to treat severe chronic pain since it has anti-nociceptive activity without developing tolerance as opposed to morphine and other opioids [93]. This fact represents a considerable advantage, especially in long-term therapy [92]. Many other drugs have been identified from marine invertebrate extracts. They have the function of inhibiting enzymes or modulating CNS channels, which are well known to be involved in developing the neurodegenerative condition. For example, potent inhibitors of cholinesterases were isolated from algae, sponges, cnidarians, mollusks, bryozoans, echinoderms, and tunicates [94]. Other drugs affecting enzymes or modulating channels are reported in Table 2. They are just a few of the bioactive compounds isolated from marine invertebrates, even if they are enough to demonstrate that many products may be considered hypothetical candidates for drug development.

Beta-secretase 1 inhibitors and glycogen synthase kinase-3 inhibitors can be considered as targets for drug development since the former is involved in producing the Aβ_1–42_, and the latter plays a role in hyper-phosphorylation of tau protein and memory impairment, which are some of the key features of AD [111,112]. Cholinesterase inhibitors represent one of the actual treatments for patients affected by a mild and moderate AD; however, they cannot arrest the disease’s progression; their action can only reduce symptoms [113]. However, the modulation of CNS voltage-dependent ion channels and CNS receptors resulted in being possible. For instance, conotoxins are antagonist of these receptors, which are known to be implicated in many physiological processes, such as memory, attention, and learning. Since their dysfunction has been related to the onset of neurological disorders, they have also been considered a potential target for treating neurodevelopmental disorders, neuropathic pain, and neurodegenerative diseases [114]. The marine cembranoids, such as lophotoxins, show antagonist activity on nicotinic acetyl-choline receptor antagonists [107]. Moreover, the activity of glycine receptors can be modulated by bioactive compounds extracted from sponges [108].

## 4. Marine Drugs in Parkinson’s Disease

PD is the most common neurodegenerative disorder, affecting 1% of the population over 65 [115]. Clinically, most patients present with a motor disorder and suffer from bradykinesia, resting tremor, rigidity, and postural instability. Other manifestations include behavioral, cognitive, and autonomic disturbances. As previously mentioned, PD is a multifactorial disease. The main causes beyond aging are oxidative stress, mitochondrial dysfunction, and the consequent loss of dopaminergic neurons; for this reason, the candidate molecules to be used as a possible protective therapy in PD should have antioxidant and anti-apoptotic activities; affecting the PI3K/Akt pathway, as well as downstream signaling.

### 4.1. Fucoidan

One molecule showing these properties, is the fucoidan [53], a polysaccharide (Figure 1) extracted from the brown algae *Saccharina japonica* (sugar content, 48%, fucose content 28%, sulfate content 29%).

Notably, fucoidan exhibited a protective effect in 1-methyl-4-phenyl-1,2,3,6-tetrahydropyridine (MPTP) C57BL/6 mice. In the study by Luo and colleagues [116], fucoidan treatment significantly improved the motor impairment in a MPTP mice model of PD. It was also able to counteract the depletion of striatal DA and reduced tyrosine hydroxylases-positive neurons in the *substantia nigra pars compacta* [116]. In an in vitro model of PD, using the 1-methyl-4-phenylpyridinium-induced MN9D dopaminergic cell line, fucoidan pre-treatment preserved cell morphology, increased mitochondrial activity, and reduced 1-methyl-4-phenylpyridinium-induced lactate dehydrogenase release; however, in line with other marine compounds, the pharmacological mechanisms underlying this protective effect is still unknown. Jhamandas et al. demonstrated that fucoidan’s neuroprotection depends on its antioxidant effect observed in rat cholinergic neurons of the basal forebrain treated with Aβ by blocking the generation of ROS. Luo et al. [116], confirmed that the neuroprotective mechanism of fucoidan might depend on its antioxidant activity even if at low concentration showed no significant effects on the other parameters. These data indicated that the change in antioxidant status contributes to the protective effect of fucoidan from MPTP-induced loss of dopaminergic neurons. Still, it is not the only mechanism exerted by this compound. Indeed, other mechanisms involved may concern the anti-inflammatory action of fucoidan.

### 4.2. Xyloketal B

A study by Nakamura et al., 2006, showed that the modulation of elements in the inflammatory pathway could regulate neurons’ state. A molecule that exhibits anti-oxidant, anti-inflammatory, and anti-apoptotic activities is the Xyloketal B (Figure 2) obtained by mangrove fungus of the South China Sea coast.

Lin et al. [117] isolated a class of compounds in 2001 (xyloketals A-G); among these compounds, xyloketal B exhibited a scavenger effect for free oxygen radicals [118] and prevented neuronal cell damage. The first to examine the protective effect of xyloketal B was Chen with his research group [119]; in particular, for testing this molecule, they used human umbilical vein endothelial cells (HUVECs) and mimicked endothelial injury with oxidized low-density lipoprotein (oxLDL). The stress with oxLDL cause cell morphological changes and decreased cell viability in this in vitro model, while the xyloketal B treatment significantly reverted these effects. This research group studied oxLDL and xyloketal B’s effect on NADPH oxidase activity (an enzyme with a key role in ROS production) to understand this protective mechanism. Interestingly, oxLDL increased the NADPH oxidase activity [120]. At the same time, the pre-treatment with xyloketal B was able to significantly lower both oxLDL-induced superoxide anion production and the mRNA expression of NADPH oxidase subunits gp91phox and p47phox [119]. These findings indicate that xyloketal B counteract ROS production via inhibiting NADPH oxidase activity and decreasing its subunits’ mRNA expression. In 2009, Zhao et al. [118] studied the neuroprotective potential of xyloketal B on an in vitro model of PC12 cell line exposed to oxygen and glucose deprivation. Initially, they tested the effect of xyloketal B pre-treatment after oxygen and glucose deprivation on cell viability, and after that, they evaluated the protective effect on mitochondria with MitoSOX dosage. The MTT assay reported significantly reduced stress with oxygen and glucose deprivation the number of viable cells, while the pre-treatment with xyloketal B counteracted this effect. For mitochondrial activity, disturbance in cellular respiration due to the glucose deficiency during ischemia causes nicotinamide adenine dinucleotide(H) accumulation in the mitochondria and ROS overproduction, which further leads to mitochondrial damage indicated by the reduction in mitochondrial membrane potential and cytochrome c release. In this study [118], mitochondrial ROS production resulted strongly enhanced upon oxygen and glucose deprivation by MitoSOX assay, while upon xyloketal B showed the MitoSOX signal intensity reduced, thus the mitochondria may represent a potential target in the anti-apoptotic effect of xyloketal B in neuronal cells.

### 4.3. Seaweeds

Another source of marine compounds with antioxidant activity is seaweeds, which have been the target of numerous studies (extensively reviewed in [121]), standing out as major producers of bioactive molecules with high antioxidant ability. Another research group tested the effects of different seaweed extracts (*S. muticum.*, *S. polyschides.*, *P. pavonica*) in SH-SY5Y cells stressed with high concentration (1M) of 6-OHDA for 24 h (cell viability reduced more than 45%). The presence of the different extracts substantially increased the cell viability, counteracting DA’s neurotoxicity. These results suggested that the extracts exerted an antiapoptotic mechanism, as observed by the mitochondrial membrane potential analyzed by JC-1 assay parallel with the inhibition of caspase-3 activity. This protective effect may be due to the antioxidant capacity of the seaweeds tested; in particular, regarding the treatments with *Ascophyllum nodosum*, *S. muticum*, and *S. polyschides*, this positive effect may be due to phlorotannins (Figure 3), exclusively found in brown seaweeds, which are characterized by high antioxidant activities.

One promising seaweed with neuroprotective potential is the *Codium Tomentosum* that showed antioxidative and antigenotoxic capacity [122]. Valentao et al. [123] tested its ability to scavenge the reactive oxygen and nitrogen species and characterized this seaweed’s chemical composition collected from the Atlantic Ocean by high-pressure liquid chromatography analysis. This species showed different organic acid, such as oxalic acid, aconitic acid, ketoglutaric acid, pyruvic acid, malic acid, malonic and fumaric acids, and a great variety of volatile compounds, such as phenolic compounds, secondary metabolites with multiple biological activities, as the plant defense mechanism under different environmental stress conditions.

### 4.4. Astaxanthin

Another promising class of molecules with therapeutic potential versus PD are carotenoids. Most carotenoids are tetraterpenoid compounds consisting of eight isoprene units and are responsible for the red, orange, and yellow colors of archaea and fungi, algae, plants. In the marine environment, these molecules are obtained by macroalgae, bacteria, and unicellular phytoplankton and play numerous and important functions, including protect chlorophyll via absorbing light energy and scavenging free radicals of oxygen [124]. Animals cannot synthesize carotenoids de novo and need to ingest carotenoids via supplementation or diet. Aquatic animals swallow carotenoids from foods, such as algae and other animals, and convert their structure via metabolic reactions leading to structural diversity. Carotenoids have been intensely studied for their role in human health, not only as natural antioxidants but also as pharmaceuticals because they generally promote health, as in cancer prevention, by boosting immune function and cognitive performance antiaging effects and anti-inflammatory agents. Despite these important pharmacological activities, carotenoid use has some limitations: for example, easy degradation, low shelf life, low solubility in water, and low bioavailability. Most carotenoids are tetraterpenoid compounds consisted of a sequence of eight isoprene units. One important carotenoid from marine organisms is AXT (introduced in previous chapters). AXT (Figure 4) is a xanthophyll carotenoid produced primarily by the marine algae *Haematococcus Pluvialis*, and it has been widely studied in a broad range of clinical applications, including cardiovascular diseases, metabolic syndrome, gastric ulcers, and cancer, which share a common feature, including inflammation and/or oxidative stress [125].

AXT is already approved as a dietary supplement and commercially available, showing no significant adverse effects. Interestingly, emerging evidence suggested that biological activities for AXT’s may counteract the pathophysiology features of PD, revealing a promising therapeutic potential in the prevention or onset of symptoms in PD patients. In the study of Castelli et al. [115], the researchers evaluated AXT’s effect, in an in vitro model of SH-SY5Y stressed with H_2_O_2_. Notably, AXT showed a cytoprotective effect, preventing or modulating the severity of neuronal death following oxidative stress injury. The authors suggested that the mechanism of action involves the modulation of neuroprotective markers, including neurotrophic pathways, i.e., BDNF and antioxidant enzymes (i.e., manganese superoxide dismutase, catalase). In another study, Grimming and collaborators [126] evaluated AXT’s effect on an in vivo model of PD. The researchers explored the capacity of a dietary pre-treatment of AXT (mice consumed an AXT-enriched diet formulated to deliver a dose of 3 mg/kg/day) in protecting against neuronal damage caused by the neurotoxin MPTP. These results indicated that the AXT-enriched diet, attenuated the loss of dopaminergic neurons induced by MPTP. The protective effects were related to a decrease in inflammation and oxidative stress. In agreement, in another investigation, the administration of AXT could block some inflammatory sequelae of the lipopolysaccharide injections and inhibit NF-κB translocation to the nucleus, thereby down-regulating tumor necrosis factor-α expression. Furthermore, AXT intake was associated with a more favorable ratio of GSH, reduced GSH to oxidized GSH in the plasma.

## 5. Marine Drugs in Alzheimer’s Disease

AD represents the most prevalent neurodegeneration worldwide [127], characterized by a deficit in language, orientation, mood control, and cognitive and memory impairment. AD is linked to aging: most cases (90%) are initially diagnosed above 65 years of age and 2–10% of total cases are diagnosed before the age of 65. New therapeutic approaches are fundamental to counteract this disorder; indeed it is projected that more than 100 million people may be affected by AD by 2050 [128].

### 5.1. Fucoxanthin

We have already described carotenoids as a possible candidate for the treatment of PD, indicating that many marine carotenoids are reported to produce antioxidant and anti-inflammatory properties. Lin et al. [129]*,* tested the possible effects of FX (Figure 5) in a mice model of AD. Research of Pangestuti et al. [130] shown that FX improves Aβ-induced-oxidative stress in microglia cells, suggesting that FX might help AD treatment [130]. For testing the effect of FX on oxidative stress and inflammation, researchers used Aβ42 induced BV2 microglia cells, demonstrating that FX attenuated pro-inflammatory secretion in BV2 cells and inhibited free radical-induced DNA oxidation in BV2 cells.

This effect was associated with a reduction of intracellular ROS formation and antioxidative enzyme induction. These results indicate that FX might be protective for neuronal cells against neurotoxic mediators released by microglia by a negative feedback regulating inflammation and oxidative stress in BV2 cells. Another interesting study by Lin et al. [129] evaluated the effects of FX on scopolamine-induced cognitive disturbances in AD model mice and investigated that this compound could inhibit the acetylcholinesterase in vitro model. The results showed that FX directly inhibits acetylcholinesterase activity in vitro by significantly reversing the acetylcholinesterase activities of scopolamine-induced alterations, suggesting that FX could directly affect enzymes in the cholinergic system. The data obtained are confirmed by the Lineweaver–Burk graphs, which suggest that FX acts as a non-competitive acetylcholinesterase inhibitor. Furthermore, Lin et al. demonstrated that FX significantly reversed the reduction in BDNF expression caused by scopolamine treatment, suggesting that FX might increase memory formation in animals. BDNF levels. In this research, FX was extracted from brown algae; the results obtained indicate that scopolamine-induced recognition disturbances in the Novel Object Recognition test, spatial learning, and memory disturbances in the Morris water maze test were indeed reversed, suggesting that FX might improve cognitive improvement, suggesting it as a valid candidate in the treatment of AD.

### 5.2. Cerebrosides

Another source of interesting marine compounds with neuroprotective activity is the sea cucumber. In particular, Li et al.’s research described the effect of cerebrosides on a rat model of AD. Sea cucumber is a traditional Asian food that has been demonstrated to contain various bioactive compounds, including cerebrosides (Figure 6) and phospholipids.

Cerebrosides are a class of neutral glycosphingolipids that are largely present in fungi, plants, animals, and marine organisms’ body wall. Moreover, cerebrosides are widely found in the brain and can be converted into ceramides; ceramides play an important role in the brain since they are further converted into sphingomyelins, sulfatides, and other glycosphingolipids, essential to maintain normal brain function and homeostasis [131]. The chemical structure of cerebrosides is unique and consists of three units: one including the monosaccharide polar head group, one the amide-linked fatty acids, and one a long-chain sphingoid bases [132]. Due to this unique structure, cerebrosides have various biological activities and, as a result, have attracted pharmaceutical research attention. In Li et al.’s study [44], an AD rat model was obtained by intraventricular injection of Aβ1–42 and treated with cerebrosides by intragastric administration. Initially, the Morris water maze results indicated that oral administration of sea cucumber cerebrosides could significantly ameliorate the impaired cognitive function in Aβ1–42-treated rats. Besides, sea cucumber cerebrosides inhibited Aβ1–42- induced apoptosis by decreasing apoptotic protein level such as Bax/Bcl-2. Moreover, sea cucumber cerebrosides improved synaptic plasticity by regulating the neurotrophic pathway BDNF/Tropomyosin receptor kinase B/cAMP Responsive Element Binding, and attenuating Aβ1–42-induced tau hyperphosphorylation by activating the PI3K/Akt/GSK3β signaling pathway. Furthermore, this research demonstrated that a diet supplemented with cerebrosides might have long-term (27 days) neuroprotective effects; the feed supplementation with sea cucumber cerebrosides could significantly alleviate the impaired cognitive function in Aβ1–42-treated rats. Some marine compounds may be used as a basis for synthesizing molecules with a more specific activity.

### 5.3. Methyl-Fascaplysin

One of the most powerful marine-derived compounds that produce in vitro neuroprotective effects is the 9-methylfascaplysin. In the research of Sun et al. [133], fascaplysin was used for designing 9-methylfascaplysin to improve its activity. Fascaplysin (Figure 7) is a extensively active benzoyl-linked β-carboline alkaloid initially extracted from the Fijian sponge *Fascaplysinopsis*; this molecule showed antitumoral and antioxidative effects [134].

The β-carbolines structure was firstly found in plants and showed pharmacological activities against neurological disease, including PD [135] and AD [136]. A previous study of Manda et al. [137] reported that fascaplysin molecule inhibits acetylcholinesterase, implying that this molecule may have positive effects on brain. Sun et al. aims to enhance the activity of fascaplysin on acetylcholinesterase using molecular modeling studies; notably, 9-methylfascaplysin resulted the greatest analog created to bind the active site of acetylcholinesterase. The authors compared 9-methylfascaplysin with fascaplysin and they concluded that 9-methylfascaplysin was more effective in inhibiting Aβ oligomerization in SH-SY5Y cell line, avoiding neuronal death at low concentrations. Molecular dynamics simulation showed that the interaction between Aβ oligomers and 9-methylfascaplysin corresponds with many hydrophobic interactions with hydrogen bonds [138]. Moreover, the initial pharmacodynamic findings revealed that these molecules can pass through blood-brain barrier and is preserved cerebrally, suggesting that it may be clinically valuable. Finally, 9-methylfascaplysin induce p-glycoprotein, as fascaplysin does [137], which suggested that fascaplysin-derivatives may be a potential multiple-target for AD.

### 5.4. Sodium Oligomannate

One of the novel marine compounds recently conditionally approved in China to treat mild to moderate AD and ameliorate brain activities is the sodium oligomannate (Figure 8).

Sodium oligomannate (GV-971) is a oligosaccharide extracted from algae developed by Shanghai Green Valley Pharmaceuticals. Sodium oligomannate was able to restore enteric microbiota, counteracting AD onset, acting on immunity system. Interestingly, the research on the enteric microbiome is growing as a potential therapeutic approach for AD [139]. Sodium oligomannate cross the blood–brain barrier thanks to type 1 glucose transporter and interact with different ligands on Aβ to prevent development of Aβ fibrils and disrupts the preformed fibrils into nontoxic monomers [140]. Sodium oligomannate showed a neuroprotective effect; indeed, it was able to counteract the Aβ toxicity in human neuroblastoma cells and exhibited helpful effects in mice models of AD, or in D-galactose-induced, or scopolamine-induced memory impairment. This molecule ameliorates cognitive performances in non-severe AD forms in a randomized, double-blind, placebo-controlled, multicenter phase III trial conducted in China. In this study [141], patients were treated for 36 weeks. A significant improvement in cognitive performances was detected beginning at 4 weeks of treatment. Now sodium oligomannate is sold as 150 mg oral capsules and the recommended dosage is three tablets twice daily.

## 6. Conclusions

Some of the drugs most in use today, such as penicillin or morphine, have been discovered thanks to the use of medicinal plants or microorganisms. For several decades, pharmaceutical companies had abandoned research on bioactive compounds to favor synthetic molecules’ research and production. After the early 2000s, research on natural compounds has returned to have its place in biomedical studies; in fact, the number of publications from 2000 to today has risen to 287 vs. 2274 in 2020 (source PubMed). One of the main reasons for this return to natural drugs is the higher biocompatibility of natural compounds than synthetic pharmaceutical products, without neglecting the great heterogeneity of these molecules and their effects. Over the past ten years, marine compounds and their derivatives have occupied more pharmaceutical and medical research space. One of the reasons is undoubtedly the large number of molecules and secondary metabolites obtainable and their variety due to the different adverse environments of the oceans and the almost infinite number of organisms that populate it (Figure 1). Among the various pharmaceutical activities on which the research focuses, there are certainly the antioxidant and anti-inflammatory properties, applicable above all to the treatment of neurodegenerative diseases. An example is represented by the carotenoids FX and AXT, starting to be used as food supplements and potential pharmacological treatments for PD and AD after years of research. In addition to the direct use of molecules synthesized by various marine organisms, pharmaceutical research uses the active compounds obtained as a basis for the design of drugs more specific than the original molecule. An example is 9-methylfascaplysin, obtained from fascaplysin, and designed to occupy the active site of acetylcholinesterase. Another important example of molecular design from marine compounds is sodium oligomannate (GV-971), an oligosaccharide obtained from a seaweed, the first to be approved as a treatment for AD in China. On this basis, it appears that marine natural products may constitute a promising “library” of natural compounds to design new treatments adjuvant to gold standard therapies, improving the efficacy of conventional drugs, and exerting synergistic or additive positive effects for neurodegenerative diseases.

## Figures and Tables

**Figure 1 marinedrugs-19-00024-f001:**
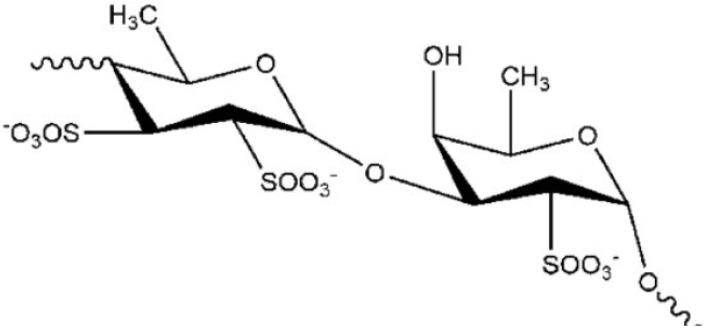
Chemical structure of fucoidan.

**Figure 2 marinedrugs-19-00024-f002:**
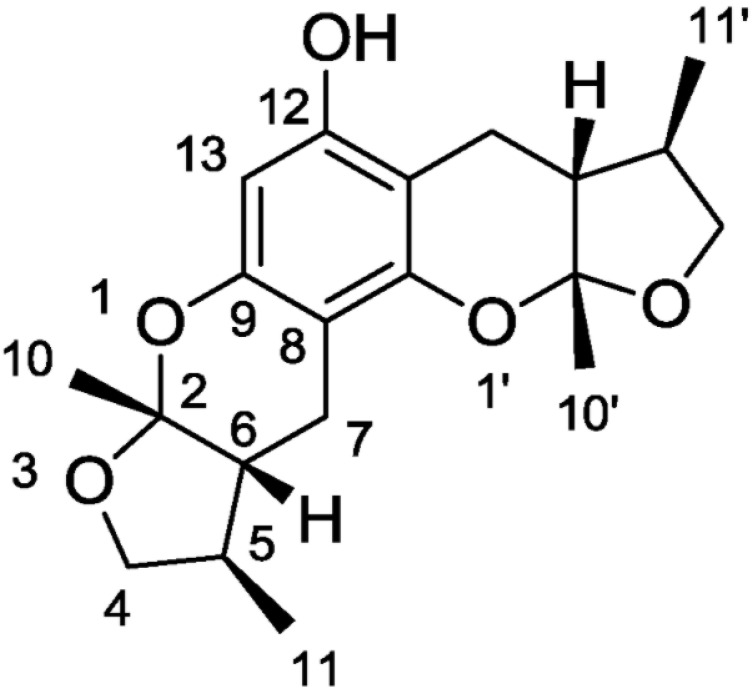
Chemical structure of xyloketal B.

**Figure 3 marinedrugs-19-00024-f003:**
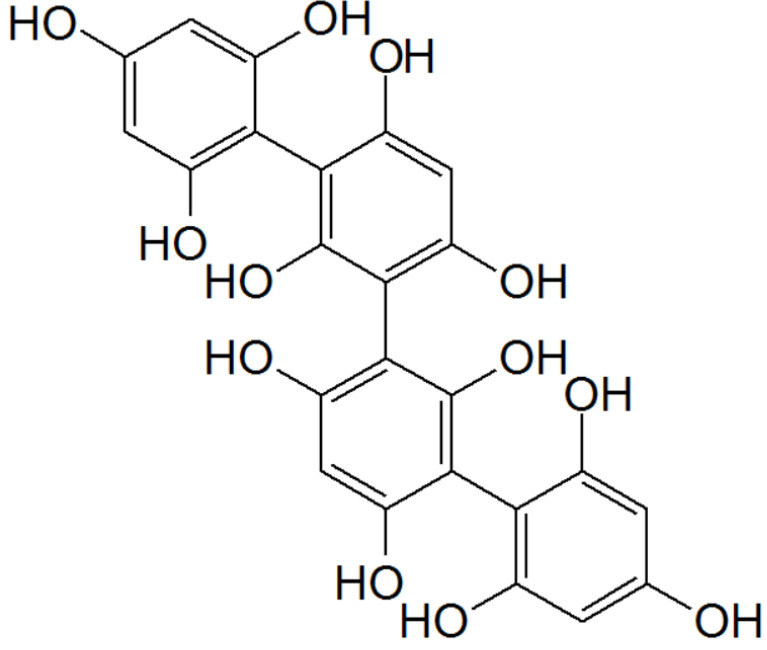
Chemical structure of tetrafucol A: a phlorotannin found in the brown algae *Ascophyllum nodosum*.

**Figure 4 marinedrugs-19-00024-f004:**
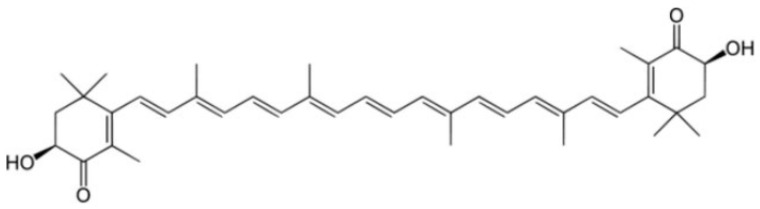
Chemical structure of AXT.

**Figure 5 marinedrugs-19-00024-f005:**
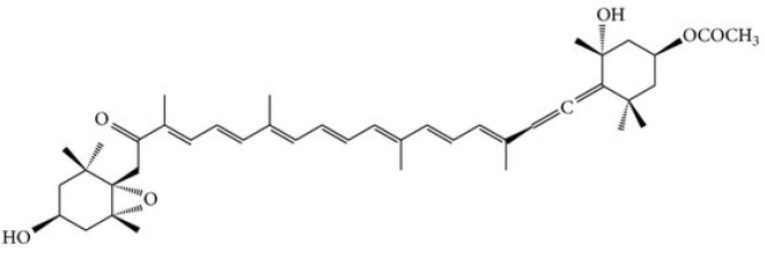
Chemical structure of FX.

**Figure 6 marinedrugs-19-00024-f006:**
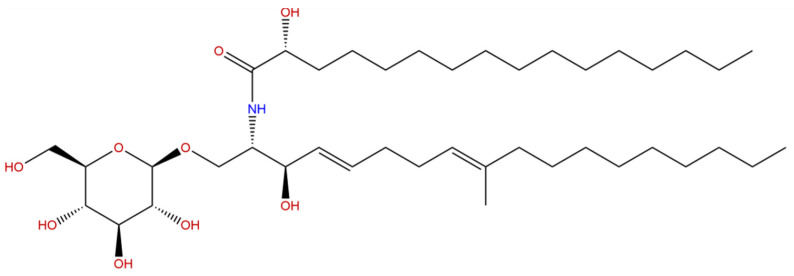
Chemical structure of Cerebrosides.

**Figure 7 marinedrugs-19-00024-f007:**
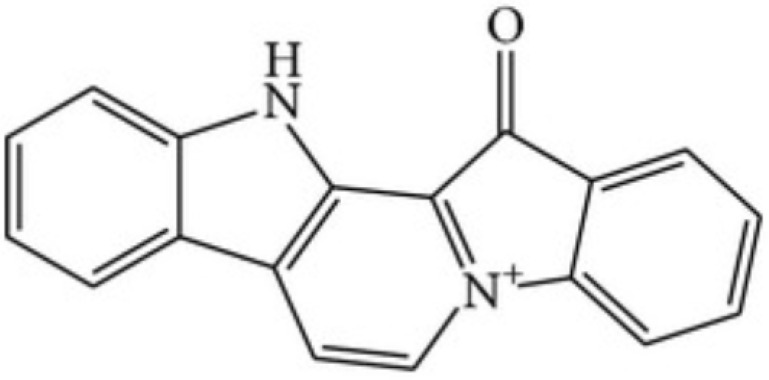
Chemical structure of fascaplysin.

**Figure 8 marinedrugs-19-00024-f008:**
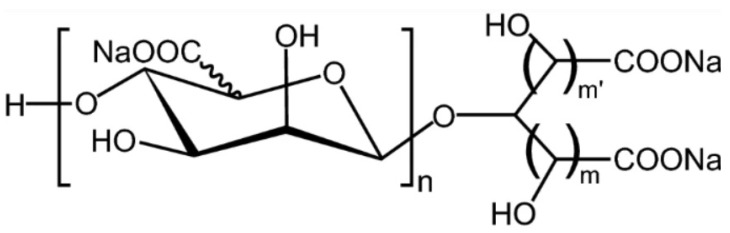
Chemical structure of sodium oligomannate.

**Table 1 marinedrugs-19-00024-t001:** Principal neuroprotective substances.

Class of Compounds	Identified Molecules of Interest	Marine Sources	Main Effects
Peptides	BPH (protein hydrolysate, consisting of active peptides Phe-Tyr-Tyr and Asp-Trp)	Lantern Fish (*Benthosema pterotum*)	Free radical scavenger, reduction of reactive species generation, and prevention of H_2_O_2_ mediated apoptosis [30]
	CFH (protein hydrolysate, consisting of 40 oligopeptides	Sea cucumber (*Cucumaria frondosa*)	In vitro: oxidative stress attenuation
			In vivo: amelioration of learning and memory deficits in D-galactose-induced aging mice [12,31]
	Conotoxins	Cone snails (genus *Conus*)	Anti-nociceptive activity and alleviation of neuropathic pain (ziconotide) [32];
			functional recovery of damaged neurons (ACV1) [33]; anti-convulsant activity (conantokin-L) [34]
	HTP-1 (H. trimaculatus-derived neuroprotective peptides Gly-Thr-Glu-Asp-Glu-Leu-AspLys)	Seahorse (*Hippocampus trimaculatus*)	In vitro: neuroprotective activity against Aβ_42_-induced apoptosis in PC12 cells [16,35]
Glycoproteins	Lectins	Green algae (*Caulerpa Cupressoides*)	In vivo: anti-nociceptive and anti-inflammatory activity in Swiss mice [36]
Pigments	AXT	Microalgae (*Haematococcus pluvialis*) Shrimp, lobster, crustacean, krill, trout, salmon	In vitro: protection from 6-OHDA-induced apoptosis and inhibition of mitochondrial impairment in SH-SY5Y [37]
			In vivo: anti-depressant effects, anti-oxidant activity (mediated by an increasing in GSH and superoxide dismutase) [38,39]
	FX	Brown seaweed (*Undaria pinnatifida*)	Reduction of oxidative stress in rat hippocampal neurons [40];
			Increase in neuron survivals in traumatic brain injury models [41]
	Mytiloxanthin (metabolite of fucoxanthin)	Tunicates and shellfish	Scavenger of singlet oxygen [42]
Lipids	Polyunsaturated fatty acids	fish oils (cod liver oil), algae, sea cucumber, microalgae	Reduction of Aβ-amyloid toxicity, anti-aggregation properties, inhibition of Aβ_40_ and Aβ_42_ fibrillogenesis [43]
Glycolipids	Glycosphingolipids (cerebrosides)	Echinoderms (sea cucumber), porifera and mollusks	Improvement of cognitive deficiency in AD rat model [44]
Glycosaminoglycans	Heparin and Heparan sulfate	Mollusks, shrimp heads (*Litopenaeus vannamei* and *Penaeus brasiliensis*),	Reduction of neuronal cell apoptosis and pro-inflammatory cytokines, neuroprotective effect in cerebral ischemia in gerbils [45]; amelioration of brain condition after stroke [46]
		crabs (*Goniopsis cruentata* and *Ucides cordatus*), sea cucumber, ascidian (*Styela plicata*), scallop, cockle (*Cerastoderma edule*), sand dollar (*Mellita quinquiesperforata*)	
	Hyaluronic acid	Shark fins, tuna eyeballs, bivalves, mussels and codfish bones	Hyaluronic acid scaffolds with neuroprotective effects in spinal cord injury [28,47]
	Chondroitin sulfate	Shark and fish cartilage, blackmouth catshark	In vitro: protection of SH-SY5Y cells against oxidative stress [29,48]
Polysaccharides	SV2-1	*Ommastrephes bartrami*	In vitro: protection of PC12 cells from 6-OHDA-induced death; anti-oxidant activity [49]
	Fucoidan	Brown algae (*Undaria pinnatifida*)	In vitro: reduction of Aβ_1–42_- and hydrogen peroxide-mediated cytotoxicity in PC12 cells [50]
	Chitosan and its derivatives	Crustaceans (shrimps and crabs)	Neuroprotective effects on peripheral nerves and Schwann cells [51]
	Carrageenan	Red algae (*Hypnea musciformis*)	In vitro: anti-oxidant and cytoprotective effects against 6-OHDA-induced neurotoxicity in SH-5YSY models [52]
	Sulfated polysaccharides	Sea weeds (*Ecklonia maxima, Gelidium pristoides, Ulva lactuca, Ulva rigida and Gracilaria gracilis*)	In vitro: stimulation of anti-oxidant activities (increase in anti-oxidant enzymes and glutathione content) in hippocampal cell line with Zn-induce damage [53]
Macrolides	Bryostatin	Brown *bryozoa* (*Bugula neritina*)	Potent modulation of protein kinase C; induction of synaptogenesis and amelioration of deficits in rats and mice models of neurodegenerative diseases [54]
	11-dehydrosinulariolide	Soft coral (*Sinularia flexibilis)*	In vitro: anti-apoptotic and anti-inflammatory activity on SH-SY5Y cells treated with 6-OHDA [55].
			In vivo: amelioration of PD symptoms in rat and zebrafish models [56]
Polycyclic ethers	Gambierol	*Gambierdiscus toxicus*	In vitro: decrease in intra- and extra-cellular levels of Aβ deposits and in tau hyperphosphorylation in triple transgenic (3xTg-AD) mice model [57]
Guanidine neurotoxins	Tetrodotoxin	*Tetraodontiformes.* (pufferfish)	Beneficial effects on acute [58], inflammatory [59] and neuropathic [60] pain
Indole alkaloids	Bromotriptamines	*Bryozoa*	In vitro: in *Xenopus.* oocytes, they act as positive allosteric modulator for two subtypes of nicotinic acetylcholine receptors (α4β2 and α2β2). They can attenuate the inhibition of Aβ_1–42_ on these receptors [61]

Abbreviations: AXT, Astaxanthin; FX, fucoxanthin; 6-OHDA, 6-Hydroxydopamine.

**Table 2 marinedrugs-19-00024-t002:** Examples of marine drugs affecting the CNS.

Pharmacological Activity	Compounds	Main Source
Beta-secretase 1 inhibitors	Xestosaprols	Indonesian marine sponges, genus *Xestospongia.* [95]
	Tasiamide B	Cyanobacteria [96]
Glycogen synthase kinase-3 inhibitors	Carteriosulfonic acids	*Sponges, genus Carteriospongia* [97]
	Leucettamines	Sponge Leucetta microraphis [98]
	Merdidianins	Ascidian Aplidium meridianum [99]
	Hymenialdisine	Sponges (various species) [100,101]
Cholinesterase inhibitors	4-acetoxy-plakinamine B	Sponges, genus Corticium [102]
	Petrosamine	Sponges, genus Petrosia n. [103]
	Alkylpyridine	Sponges, *Reniera sarai* [104]
	(and alkylpyridinium derivatives)	
Nicotinic acetylcholine receptor antagonists	Α-conotoxins	Sea snail, genus *Conus* species: *geographus, imperialis, vexillum, quercinum* [105,106] Octocorals [107]
	Cembranoids	
	(lophotoxin)	
Glycine receptors modulators	Ircinialactams	Australian sponges, family Irciinidae [108]
Neuronal growth inducers	Dysideamine	Indonesian marine sponge, genus *Dysidea* [109]
Neurotrophic-like agents	Linckosides	Okinawan starfish *Linckia laevigata* [110]

## Data Availability

Not applicable.

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
