# Peer review of "Benefits under the Sea: The Role of Marine Compounds in Neurodegenerative Disorders"

_marinedrugs, 2021, doi:10.3390/md19010024_

Round 1

Reviewer 1 Report

The review by M. Catanesi et al. is an interesting contribution to the field of marine natural products that could be used in treatment of neurodegenerative diseases, especially the Alzheimer and the Parkinson disease.

While the topic is appealing and some part of the manuscript are very well elaborated, some others are not easy to follow. I suggest to improve the manuscript by re-organizing it in a way explained below. Also, in my opinion, some crucial references are missing and should be added to the text. For example, I am surprised that there is no reference to the paper of Sakai and Swanson (2014) Nat Prod Rep. 31(2): 273–309, in which the neuroprotective marine products are systematically reviewed. In general, I invite the authors to search particulary the Natural Product Reports for comprehensive reviews on bioactive marine natural products.

Introduction:

While I appreciated very much the description of the therapeutic use of marine invertebrates in ancient populations, there is almost nothing about the recent history of marine natural products. Author should add this; there are several excellent reviews on this topic (e.g. Gerwick & Moore (2012) Chem Biol 19(1): 85–98; Leal et al. (2012), PLoS ONE 7(1) e30580….).

Further, marine invertebrates do not produce bioactive compounds only as a result of their exposure to atypical and extreme habitats, but also as defense molecules against predators, fouling organisms, and for predation. Please, add these facts.  

Further chapters (general remarks):

I would suggest starting (after Introduction) with the chapter Neurodegenerative diseases (now chapter 3). This chapter, in my opinion, should serve as an introduction to what follows: the marine agents that could be used in curing these diseases. As the review for the Marine Drugs, in my opinion, should focus on compounds, and not on the detailed description of the mechanism of the disease, I suggest this chapter to be shortened, or at least split in paragraphs for better comprehension. There are too many details and abbreviations also (see below “other points”).

This chapter should be followed by all the other chapters as they appear in the current text, starting with the “Classification of marine neuroprotective compounds”, followed by the Table 1. Please refer to the already mentioned Sakai and Swanson review for potentially missing compounds.

In the following chapter (Principal methods….) please be careful when listing the separation techniques (“chromatography, reverse phase columns, fractionation”): all this refers to chromatographic separation. There are some repetitions in the first half of the text; please shorten.

Please change the title of the chapter 2.3 to: “The effects of marine compounds on the central nervous system”.

Line 250: when specifying the cholinesterase inhibitors from marine sources, it would be much more appropriate to cite the review by Moodie et al. (2019), Nat Prod Rep 36, 1053–1092.

Table 2: please consult Sakai and Swanson review for potentially missing important compounds.

Current chapters 3.1 and 3.2.: these is the most important part of the paper, and I really miss better structuring of the text; e.g. each new compound or a class of compounds should start with the new paragraph and should be accompanied by the structural formula of the compound. The Figure 1, that appears at the end of the text, is suitable for the graphical abstract and not for the main text.

Line 562: which kind of long-chain base do you have in mind; the sphingiod base? Please, define.

Other points:

The Latin terms like »in vivo«, »in vitro«, »de novo«, »via«, »substantia nigra«….should be written in Italics. Also, several genera and species names are not italicized. Please correct. Please note that Conus Magnum should be Conus magus. Please, add that Saccharina japonica is the brown alga.

There are really too many abbreviations in the text, making it hard to follow in some parts. Please, do not abbreviate all the words; e.g. do not abbreviate the ones that appear less than 5 times in the text. The text will be much more user friendly in that way. Further, please define the abbreviation the FIRST time it appears in the text. For example, there are several abbreviations in the Table 1 which were not defined before. Please, define them at least in the table legend.

The names of several natural compounds and enzymes are often capitalized in the text, even if they do not appear at the beginning of the sentence. Please, correct: they should not be capitalized. Metilfascaplysina should be written as methylfascaplysin (in two occasions).

Author Response

Reviewer 1

Comments:

The review by M. Catanesi et al. is an interesting contribution to the field of marine natural products that could be used in treatment of neurodegenerative diseases, especially the Alzheimer and the Parkinson disease.

While the topic is appealing and some part of the manuscript are very well elaborated, some others are not easy to follow. I suggest to improve the manuscript by re-organizing it in a way explained below. Also, in my opinion, some crucial references are missing and should be added to the text. For example, I am surprised that there is no reference to the paper of Sakai and Swanson (2014) Nat Prod Rep. 31(2): 273–309, in which the neuroprotective marine products are systematically reviewed. In general, I invite the authors to search particulary the Natural Product Reports for comprehensive reviews on bioactive marine natural products.

Response: We would like to thank the Reviewer 1 for the time spent in revising our manuscript and for the valuable comments provided that helped in improving our review article. We tried to address all the points raised. Thank you for the observation, the paper of Sakai and Swanson (2014) was added in the line 363 and on the table 2 as suggested.

Introduction:

While I appreciated very much the description of the therapeutic use of marine invertebrates in ancient populations, there is almost nothing about the recent history of marine natural products. Author should add this; there are several excellent reviews on this topic (e.g. Gerwick & Moore (2012) Chem Biol 19(1): 85–98; Leal et al. (2012), PLoS ONE 7(1) e30580….).

Response:  We appreciate the suggestion and we now added the part regarding the recent history of marine drugs in the lines 57 to 65 as suggested by the Reviewer.

Further, marine invertebrates do not produce bioactive compounds only as a result of their exposure to atypical and extreme habitats, but also as defense molecules against predators, fouling organisms, and for predation. Please, add these facts.  

Response: Thank you for the comment, the information about molecules produced by marine invertebrates against predator, fouling organisms and for predation was added in the lines 67 to 86.

I would suggest starting (after Introduction) with the chapter Neurodegenerative diseases (now chapter 3). This chapter, in my opinion, should serve as an introduction to what follows: the marine agents that could be used in curing these diseases. As the review for the Marine Drugs, in my opinion, should focus on compounds, and not on the detailed description of the mechanism of the disease, I suggest this chapter to be shortened, or at least split in paragraphs for better comprehension. There are too many details and abbreviations also (see below “other points”). This chapter should be followed by all the other chapters as they appear in the current text, starting with the “Classification of marine neuroprotective compounds”, followed by the Table 1. Please refer to the already mentioned Sakai and Swanson review for potentially missing compounds.

Response: We thank the Reviewer 1 for the suggestions. The chapter “Neurodegenerative diseases” was moved as suggested (chapter 2) and shortened, avoiding the number of the abbreviation.

In the following chapter (Principal methods….) please be careful when listing the separation techniques (“chromatography, reverse phase columns, fractionation”): all this refers to chromatographic separation. There are some repetitions in the first half of the text; please shorten.

Response: We apologize for the oversights, the chapter 2.1 (Principal Methods of Extraction, Separation, Isolation, and Identification) was modified according with the Reviewer’s suggestions.

Please change the title of the chapter 2.3 to: “The effects of marine compounds on the central nervous system”.

Response: Thank you for the suggestion, we now renamed the chapter 2.3.

Line 250: when specifying the cholinesterase inhibitors from marine sources, it would be much more appropriate to cite the review by Moodie et al. (2019), Nat Prod Rep 36, 1053–1092.

Response: Thanks for the suggestion, the reference indicated was added in line 334.

Table 2: please consult Sakai and Swanson review for potentially missing important compounds

Response: Thank you for the observation, we now added this information in Table 2.

Current chapters 3.1 and 3.2.: these is the most important part of the paper, and I really miss better structuring of the text; e.g. each new compound or a class of compounds should start with the new paragraph and should be accompanied by the structural formula of the compound. The Figure 1, that appears at the end of the text, is suitable for the graphical abstract and not for the main text.

Response: Thanks for the observation, we re-structured the text as suggested. We now added the structural formula for each compound and the chapters have been divided into subparagraphs. The figure 1 has been removed and added as graphical abstract.

Line 562: which kind of long-chain base do you have in mind; the sphingiod base? Please, define.

Response: Thanks for the observation, the sphingoid base was added on the line 713.

Other points:

The Latin terms like »in vivo«, »in vitro«, »de novo«, »via«, »substantia nigra« should be written in Italics. Also, several genera and species names are not italicized. Please correct. Please note that Conus Magnum should be Conus magus. Please, add that Saccharina japonica is the brown alga.

Response: Thanks for the observations, we now corrected the latin terms in Italics and added the information about Saccharina japonica.

There are really too many abbreviations in the text, making it hard to follow in some parts. Please, do not abbreviate all the words; e.g. do not abbreviate the ones that appear less than 5 times in the text. The text will be much more user friendly in that way. Further, please define the abbreviation the FIRST time it appears in the text. For example, there are several abbreviations in the Table 1 which were not defined before. Please, define them at least in the table legend.

Response: Thank you for the suggestion, some abbreviations have been removed and a table legend added.

The names of several natural compounds and enzymes are often capitalized in the text, even if they do not appear at the beginning of the sentence. Please, correct: they should not be capitalized. Metilfascaplysina should be written as methylfascaplysin (in two occasions).

Response: We apologize for the oversights, the name and capital letters have been corrected.

Reviewer 2 Report

Line 67: Please add the reference for the claim that some of these compounds have anticoagulant properties.

It is the first time I found a figure mentioned in the conclusion section. Added to this, I do not find it that important considering that you already have some well structured tables that cover the information from Figure 1. Thus, please remove Figure 1.

From this review I can observe that most compounds have antiinflammatory properties. Although it is true that neurodegenerative diseases are characterised by a proinflammatory environment I would find it unlikely that antiinflammatory compounds would have an important effect especially in advanced forms of disease. Nonetheless, some of the mentioned compounds have shown neuroregenerative properties, making them have some potential in these diseases.

Overall, it is a well structured and argumented manuscript. I consider that it would be suitable for publication after minor revisions.

Author Response

Reviewer2

Line 67: Please add the reference for the claim that some of these compounds have anticoagulant properties.

Response: We apologize for the oversight, the references (doi:10.3390/md11020399) was now added in the line 92.

It is the first time I found a figure mentioned in the conclusion section. Added to this, I do not find it that important considering that you already have some well structured tables that cover the information from Figure 1. Thus, please remove Figure 1.

Response: Thanks for the suggestion, the figure was removed from the main text and used as visual abstract.

From this review I can observe that most compounds have antiinflammatory properties. Although it is true that neurodegenerative diseases are characterised by a proinflammatory environment I would find it unlikely that antiinflammatory compounds would have an important effect especially in advanced forms of disease. Nonetheless, some of the mentioned compounds have shown neuroregenerative properties, making them have some potential in these diseases.

Overall, it is a well structured and argumented manuscript. I consider that it would be suitable for publication after minor revisions.

Response: We would like to thank the Reviewer 2 for the time spent in revising our manuscript and for the valuable comments provided that helped in improving our review article. We tried to address all the points raised.

Round 2

Reviewer 1 Report

The authors have amended the appear according to reviewer’s suggestion, and I find it much more comprehensive in the revised form. I suggest that the paper is accepted after some additional minor points, as suggested below, has been fixed.

Minor points:

1. Please, carefully check the text once again and do not capitalize the names of compounds, receptors, channels, enzymes etc. There are many of these situations in the text, e.g. “CNS Voltage-Dependent Ion Channels and CNS Re336 Receptors”, “Beta-secretase 1 inhibitors and Glycogen synthase”, “Chemical structure of Fucoidan”, “Chemical structure of Xyloketal B”, “The Authors compared 9-methylfascaplysin”….and many others.

2. Also, please carefully check all the abbreviations. Some of them are still not defined, or are not necessary. Example: »….pre-treatment preserved cell morphology, in369 creased mitochondrial activity, and reduced MPP-induced LDH release….«

3. Please, rewrite the following phrase: “Another class of marine compounds with antioxidant activity is seaweeds, which been the target of numerous studies, standing out as major producers of bioactive molecules with high antioxidant ability”. Seaweeds are not a class of compounds, but a source of compounds with antioxidant activity.

Author Response

The authors have amended the appear according to reviewer’s suggestion, and I find it much more comprehensive in the revised form. I suggest that the paper is accepted after some additional minor points, as suggested below, has been fixed.

RESPONSE: We appreciate your positive comments. We tried to address also these minor points.

Minor points:

  1. Please, carefully check the text once again and do not capitalize the names of compounds, receptors, channels, enzymes etc. There are many of these situations in the text, e.g. “CNS Voltage-Dependent Ion Channels and CNS Re336 Receptors”, “Beta-secretase 1 inhibitors and Glycogen synthase”, “Chemical structure of Fucoidan”, “Chemical structure of Xyloketal B”, “The Authors compared 9-methylfascaplysin”….and many others.

RESPONSE: We apologize for the oversights and we now carefully proofread the manuscript.

2. Also, please carefully check all the abbreviations. Some of them are still not defined, or are not necessary. Example: »….pre-treatment preserved cell morphology, in369 creased mitochondrial activity, and reduced MPP-induced LDH release….«

RESPONSE: We thank the reviewer for the suggestion and we now carefully proofread the manuscript.

3. Please, rewrite the following phrase: “Another class of marine compounds with antioxidant activity is seaweeds, which been the target of numerous studies, standing out as major producers of bioactive molecules with high antioxidant ability”. Seaweeds are not a class of compounds, but a source of compounds with antioxidant activity.

RESPONSE: We thank the reviewer for highlighting this point. We now rewrote the sentence as suggested.